# Influence of Cholesterol on the Orientation of the Farnesylated GTP-Bound KRas-4B Binding with Anionic Model Membranes

**DOI:** 10.3390/membranes10110364

**Published:** 2020-11-22

**Authors:** Huixia Lu, Jordi Martí

**Affiliations:** Department of Physics, Technical University of Catalonia-Barcelona Tech, 08034 Barcelona, Spain; huixia.lu@upc.edu

**Keywords:** KRas-4B, mutation, post-translational modification, HVR, anionic plasma membrane, signaling, cholesterol

## Abstract

The Ras family of proteins is tethered to the inner leaflet of the cell membranes which plays an essential role in signal transduction pathways that promote cellular proliferation, survival, growth, and differentiation. KRas-4B, the most mutated Ras isoform in different cancers, has been under extensive study for more than two decades. Here we have focused our interest on the influence of cholesterol on the orientations that KRas-4B adopts with respect to the plane of the anionic model membranes. How cholesterol in the bilayer might modulate preferences for specific orientation states is far from clear. Herein, after analyzing data from in total 4000 ns-long molecular dynamics (MD) simulations for four KRas-4B systems, properties such as the area per lipid and thickness of the membrane as well as selected radial distribution functions, penetration of different moieties of KRas-4B, and internal conformational fluctuations of flexible moieties in KRas-4B have been calculated. It has been shown that high cholesterol content in the plasma membrane (PM) favors one orientation state (OS${}_{1}$), exposing the effector-binding loop for signal transduction in the cell from the atomic level. We confirm that high cholesterol in the PM helps KRas-4B mutant stay in its constitutively active state, which suggests that high cholesterol intake can increase mortality and may promote cancer progression for cancer patients. We propose that during the treatment of KRas-4B-related cancers, reducing the cholesterol level in the PM and sustaining cancer progression by controlling the plasma cholesterol intake might be taken into account in anti-cancer therapies.

## 1. Introduction

The cell membrane plays an important role in controlling the passing of nutrients, wastes, drugs, and heat between the inner and outer parts of a cell. The principal components of human cellular membranes are phospholipids, cholesterol, and proteins, etc. Moreover, the concentration of each constituent differs for different types of cells. Phospholipids provide the framework to biomembranes and they consist of two leaflets of amphiphilic lipids with a hydrophilic head and one or two hydrophobic tails which self-assemble due to the hydrophobic effect [1,2]. For instance, 1,2-dioleoyl-sn-glycero-3-phosphocholine (DOPC) belongs to the unsaturated phospholipids which is a typical constituent of real biological membranes. Furthermore, 1,2-dioleoyl-sn-glycero-3-phospho-L-serine (DOPS) is the most common anionic lipid in the plasma membrane (PM) of mammalian cells, which is preferentially targeted by the PM intracellular surface protein KRas-4B [3] for signal transduction.

PM systems have been extensively studied over several decades [4,5,6,7,8] on their association with proteins. Recent studies have shown that the role of proteins and their interactions with components of PM is extremely important to understand the mechanisms of protein anchoring into the membrane that can lead to oncogenesis [9]. GTPases are a large family of hydrolase enzymes that bind to the nucleotide guanosine triphosphate (GTP) and hydrolyze it to guanosine diphosphate (GDP). GDP/GTP cycling is controlled by two main classes of regulatory proteins. Guanine-nucleotide-exchange factors (GEFs) promote the formation of the active, GTP-bound form, while GTPase-activating proteins (GAPs) inactivate Ras by enhancing the intrinsic GTPase activity to promote the formation of the inactive GDP-bound form [10,11,12]. Ras proteins are small molecular weight GTPases and function as GDP/GTP-regulated molecular switches controlling pathways involved in critical cellular functions, like cell proliferation, signaling, cell growth, and anti-apoptosis pathways [13]. The three Ras genes give rise to three base protein sequences: KRas, HRas, and NRas. Over 30% of cancers are driven by mutant Ras proteins, thereinto, one method called catalog of somatic mutations in cancer (COSMIC) [14] confirms that HRas (3%) are the least frequently mutated Ras isoforms in human cancers, where KRas (86%) are the predominantly mutated isoforms followed by NRas (11%) [15].

KRas can be found as two splice variants designated KRas-4A and KRas-4B. They both have polybasic sequences that facilitate membrane-association in acidic membrane regions [16], however, for KRas-4A, it is covalently modified by a single palmitic acid. KRas-4B is distinguished from KRas-4A isoform in the residue 181 that serves as a phosphorylation site within its flexible hypervariable region (HVR, residues 167–185) that contains the farnesyl group (FAR) serving as the lipid anchor. The HVR of KRas-4B contains multiple amino-acid lysines that act as an electrostatic farnesylated switch which guarantees KRas-4B’s association with the negatively charged phospholipids in the inner PM leaflet. It has been reported that the KRas-4B activation level in diseased cells is linked to phosphatidylserine contents [17]. Anionic lipids could influence the membrane potential which in turn regulates the orientation, location, and signaling ability of KRas-4B [18,19].

The catalytic domain (CD, residues 1–166), which contains the catalytic lobe (lobe 1, residues 1–86) and the allosteric lobe (lobe 2, residues 87–166), highly homologous, conserved, and the structure is shared and identical for KRas-4A and KRas-4B. According to P. Prakash et al. [20,21,22], three distinct orientation states of the oncogenic G12V-KRas-4B mutant on the membrane have been reported, namely, OS${}_{1}$, OS${}_{2}$, and OS${}_{0}$. OS${}_{1}$, with an accessible effector-binding loop, and OS${}_{2}$, with the effector-binding loop occluded by the membrane, has been reported. They differ in the accessibility of functionally critical switch loops to the downstream effectors, suggesting that membrane reorientation of KRas-4B on the inner cell leaflet may modulate its signaling [21]. The idea of the more flexible in the structure of proteins, the larger the number of their populated states have been pointed out [23]. We and other researchers have recently shown that despite the HVR and FAR anchor, the CD of KRas-4B could interact with anionic model membranes by forming steady salt bridges and hydrogen bonds to help organize its orientations in cells [18,24,25].

All Ras proteins’ signaling strongly depends on their correct localization in the cell membrane and it is essential for activating downstream signaling pathways. KRas-4B function, membrane association and interaction with other proteins are regulated by post-translational modifications (PTMs) [26,27,28], including ubiquitination, acetylation, prenylation, phosphorylation, and carboxymethylation, see Figure 1. Firstly, the prenylation reaction, catalyzed by cytosolic farnesyltrasferase (FTase) or geranylgeranyltransferase (GGTase), proceeds through the addition of an isoprenyl group to the Cys-185 side chain. Then, farnesylated KRas-4B is ready for further processing: hydrolysis, catalyzed by the endopeptidase enzyme called Ras-converting enzyme 1 (RCE1); during the process, the VIM motif (HVR tail is composed of three amino acids: valine-isoleucine-methionine) of the C-terminal Cys-185 is lost in step 2.

In step 3, KRas-4B is transferred to the endoplasmic reticulum for carboxymethylation at the carboxyl terminus of Cys-185 catalyzed by isoprenylcysteine carboxyl methyltransferase (ICMT), forming a reversible ester bond. The reversible ester bond can go through decarboxymethylation, catalyzed by prenylated/polyisoprenylated methylated protein methyl esterases (PMPEases) giving rise to a farnesylated and demethylated KRas-4B (KRas-4B-Far). Carboxymethylation is one of the best known reversible PTMs in HVR [29]. This reversible reaction can modulate the equilibrium of methylated/demethylated KRas-4B population (KRas-4B-FMe/KRas-4B-Far) in tumors and consequently can impact downstream signaling, protein–protein interactions, or protein–lipid interactions [30]. Another well-known reversible PTM in the HVR is phosphorylation [28,31,32]. There are two sites (Ser-171 and Ser-181) within HVR that could be phosphorylated. Phosphorylation involves the addition of phosphate (PO${}_{4}^{3-}$) group to the side chain of the amino acid serine, then the phosphorylated serine is obtained. In this work, we have only applied the phosphorylation at Ser-181 (PHOS) for the oncogenic KRas-4B. Phosphorylation at Ser-181 operates a farnesyl-electrostatic switch that reduces but does not completely inhibit membrane association and clustering of KRas-4B, leading to the redistribution of the cytoplasm and endomembranes [27,33,34]. Functionally, the phosphorylation of KRas-4B can have either a negative [35,36] or positive [34,37] regulatory effect on tumor cell growth, depending on the conditions [30]. For instance, from a molecular dynamics (MD) simulation of the HVR peptide with the FAR at Cys-185 of KRas-4B in two types of model membranes, it has been observed that phosphorylation at Ser-181 prohibits spontaneous FAR membrane insertion [38]. According to Agell et al., KRas-4B binding with calmodulin leads to different behaviors: short or prolonged signaling whether KRas-4B is at its phosphorylated state on residue Ser-181 [34,39]. Moreover, according to Barcelo et al. [37], phosphorylation at Ser-181 of oncogenic KRas is required for tumor growth. In summary, the phosphorylation of the HVR of KRas-4B can affect its function, membrane association, and reacting with downstream effectors [30].

Phosphodiest-eraseδ (PDEδ) has been revealed to promote effective KRas-4B signaling by sequestering KRas-4B-FMe from the cytosol by binding the prenylated HVR and help to enhance its diffusion to the PM throughout the cell, where it is released to activate various signaling pathways required for the initiation and maintenance of cancer [40,41,42,43], hoping to identify a panel of novel PDEδ inhibitors. As described in our earlier work [24], despite KRas-4B-Far’s poor affinity for PDEδ [40], it can still be transferred to the PM through trapping and vesicular transport without the help of PDEδ [44]. Moreover, according to Ntai et al., 91% of the mutant KRas-4B and 51% of wild-type KRas-4B proteins in certain colorectal tumor samples have been found to exist in its KRas-4B-Far form. While there is a relatively high abundance of KRas-4B-Far (wild-type and mutant) lacking the methyl group of Cys-185 in tumors, the effects of demethylated KRas-4B-Far on downstream signaling have yet to be determined [45]. While extensive research has been focused on methylated KRas-4B-FMe, we believe that demethylated KRas-4B-Far could play a big role in the signaling pathway that happens on the inner leaflet of the membrane bilayers.

Cholesterol plays an important role in maintaining the structure of different membranes and regulating their functions [46,47], and cancer development as well [48]. In some types of cells, however, the distribution of cholesterol is different in the inner and outer leaflets of the membrane, ranging from 0.1% to 50% of total membrane lipids depending on the cell type [49]. The average value of cholesterol concentration in the PM fell in the range of the reported value of 19–40 % [50,51,52]. For example, in red cells, the percentage of cholesterol differs in the outer leaflet (51%) and the inner leaflet (49%) [53]. In other types of cells, cholesterol constitutes about 33.3% of the outer leaflet in healthy colorectal cells [50]. Previous simulations with percentages of 10%, 20%, and 40% for DPPC lipid bilayers showed no further relevant physical changes compared to the cholesterol percentages of 0%, 30% and 50% adopted in our previous work [54].

The fluidity of the membrane is mainly regulated by the amount of cholesterol, in such a way that membranes with high cholesterol contents are stiffer than those with low amounts but keeping the appropriate fluidity for allowing normal membrane functions. Extensive research has been done on the influence of cholesterol on the mechanism of membrane structures [55], the 18-kDa translocator protein (TSPO) binding in the brain [56], etc. Pancreatic ductal adenocarcinoma (PDAC) is one of the most lethal cancers with the lowest survival rate (five-year survival of only 8%) among the cancers reported by the American Cancer Society [57]. There is evidence that shows that high cholesterol increases PDAC cancer risk. According to Chen et al. [58], a linear dose–response relationship has attested that the risk of pancreatic cancer rises by 8% with 100 mg/day of cholesterol intake through the dose-response analysis. In addition, cholesterol does not influence the mortality among patients with PDAC cancer for both statin users and nonusers measured at different time windows and analyzed as continuous, dichotomous, and categorical variables [59].

The mechanisms underlying the cholesterol-cancer correlation have not been fully elucidated. In the present work, we used molecular dynamics (MD) simulations, a very successful tool to describe a wide variety of molecular setups at the all-atom level, such as complex biological and aqueous systems [60,61,62]. We have investigated whether cholesterol in membranes affects the signaling of Ras proteins by interfering with their orientations when the oncogenic and wild-type KRas binding with the membrane. Moreover, two percentages of cholesterol (0% and 30%) have been considered. Gaining a precise understanding of the influence of cholesterol on the reorientation of mutant and wild-type KRas-4B-Far binding at the anionic model membranes is the goal of the current work.

## 2. Results and Discussion

### 2.1. Area Per Lipid

Area per lipid is often used as the key parameter when assessing the validity of MD simulations of cell membranes. It has been proposed that a good test for such validation is the comparison of the area per lipid and thickness of the membrane with experimental data obtained from scattering density profiles [63]. The area per lipid and thickness along the simulation time of the last 500 ns have been computed (see Figure A1 of Appendix A.1) and the average values are reported in Table 1.

The experimental value of 0.71 nm${}^{2}$ for area per lipid of DOPC/DOPS (4:1) at 297 K was reported in Ref. [64] and the experimental value of thickness to be of 3.94 nm at 303 K was reported by Novakova et al. [65]. As temperature increases, atoms in the lipid structure oscillate more perpendicularly to their bonds. So, increasing the temperature of the system leads to an increased area per lipid of certain phosphatidylcholine lipids, as was observed for temperatures below 420 K [66]. It was previously reported that KRas-4B can interact with head groups of DOPC and DOPS lipid molecules through long-lived salt bridges and hydrogen bonds [24]. Accordingly, when the system temperature rises, for the same model membrane, its thickness decreases. Our results of the area per lipid (0.03 nm${}^{2}$) and thickness (∼0.1 nm) are smaller than the experiment values for pure lipid systems. The main reason is the contribution of the joint effect of raising the system temperature and the appearance of KRas-4B-Far, showing a slight condensing effect on the membrane. According to earlier research [67], in the case of relatively high cholesterol concentration, 10∼20% smaller area per lipid will be considered to be reasonable and close to equilibrium ones. From another work [68], compared to pure DMPC bilayer, the area per lipid of DMPC with cholesterol (30%) has been decreased by 32% from 0.62 to 0.42 nm${}^{2}$. In the regime with chol. ≤ 30%, the area per lipid has been reported to decrease sharply as cholesterol is added into the system [69]. In Table 1, the area per lipid for high cholesterol cases (chol.-30%) has been decreased by 23% when compared with the cholesterol-free cases (chol.-0%). The results make much sense when compared with the experimental values confirming cholesterol’s condensation effect on DOPC/DOPS membrane bilayers. The results of the area per lipid and thickness of membrane bilayers we have investigated are in good agreement with experimental values. Hence, the validity of MD simulations reported here, regarding the structural characteristics of the membrane, has been established.

### 2.2. Preferential Localization of Kras-4b-Far on Membranes

Ras proteins are activated following an incoming signal from their upstream regulators and interact with their downstream effectors only when they are anchored into the membrane and being at the GTP-bound state. Tracking the movement of the FAR of KRas-4B-Far and GTP along the membrane normal could give us direct information on how the KRas-4B proteins and GTP molecule regulate each other. We report in Figure 2, the *Z*-axis positions of the centers of FAR and GTP from the center of lipids (i.e., *Z* = 0) using the second half of 1000 ns simulation for all cases.

As is described in Ref. [24], the FAR of the wild-type KRas-4B-Far is revealed to be able to anchor into and depart from the membrane without difficulty in the chol.-30% case when GTP favors bind with the interface of the membrane through salt bridges and hydrogen bonds, located at around 2.39 nm from the membrane center. The FAR can have two preferred localisations: (1) 3.90 nm when FAR wanders in the water region, and (2) 1.73 nm when FAR anchors inside of the PM. However, in the chol.-free case, the FAR of the wild-type KRas-4B-Far is found to be anchoring constantly into the anionic membrane for the entire duration of the simulations. FAR keeps locating around 1.30 nm, while GTP keeps binding to the CD, staying around 4.38 nm.

For the mutant KRas-4B-Far, when diffusing in the DOPC/DOPS (4:1) bilayer, GTP tends to wander around 2.78 nm away from the membrane center along with the membrane normal direction. When 30% of cholesterol was considered, GTP favors binding with the CD instead of wandering near the interface region of the membrane. For both oncogenic cases, FAR is revealed to be anchoring constantly into the anionic membrane as a function of the simulation time, as might be expected.

By comparing the four systems we studied, we propose that adding cholesterol into the system has less influence on the behavior of FAR of the oncogenic KRas-4B-Far anchoring to the anionic membrane. Moreover, for a different type of KRas-4B-Far, GTP’s localization cannot be predicted according to different types of mutations in the KRas-4B’s structure and the constitution of the cell membrane we are studying. Remarkably, the existence of cholesterol helps FAR of the mutant KRas-4B-Far anchor 0.17 nm deeper into the anionic membrane than the chol.-0% case.

### 2.3. Conformation of the 5-Aa-Sequence in The Hvr

As suggested by Dharmaiah et al. [40], a 5-amino-acid-long sequence motif in its HVR (K-S-K-T-K, residues 180-184), which is shared by KRas-4B-Far and KRas-4B-FMe, may enable PDEδ to bind prenylated KRas-4B.

The root mean square deviation (RMSD) of certain atoms in a molecule with respect to a reference structure, defined as Equation (Equation 1), is the most commonly used quantitative measure of the similarity between two superimposed atomic coordinates [70].
(1)$$RMSD=\sqrt{\frac{1}{n}\sum_{i=1}^{n}d_{i}^{2}}$$
where *d*${}_{i}$ is the distance between the two atoms in the i-th pair and the averaging is performed over the *n* pairs of atoms.

A closer investigation of RMSD of this 5-aa-sequence of the HVR of two KRas-4B-Far (wt. and onc.) binding to two different anionic model membranes has been done.

Figure 3 presents the results of adding cholesterol into the system to their respective reference structures. Obviously, cholesterol doesn’t have as much impact as two mutations (G12D and PHOS) in its sequence for the same type of KRas-4B-Far, highlighting the significant influence of the mutations on the conformational change of the 5-aa-sequence in the HVR. It also demonstrates that for the wild-type KRas-4B-Far protein, the RMSD of the 5-aa-sequence ranges from 0.24 to 0.3 nm, and for the oncogenic one, the value ranges from 0.35 to 0.42, due to inherent structural flexibility.

### 2.4. Orientational Distributions of Kras-4b-Far on Different Anionic Membranes

Several previous studies have shown that the orientations of Ras proteins on membranes significantly impact their function in cell [20,21,71,72,73]. Cell membranes are platforms for cellular signal transduction. Their structure and function depend on the composition of cholesterol and related phospholipids [74]. Furthermore, both clinical and experimental studies have found that hypercholesterolemia and a high-fat high-cholesterol diet can affect cancer development [48,75]. Increased cholesterol levels in the human body are associated with a higher cancer incidence, and reducing its level through drugs (for instance: statins) could reduce the risk and mortality of some cancers, such as prostate, colorectal, and breast cancer [76,77,78]. Increased serum cholesterol levels could be used as an indicator for developing cancers, such as colon, prostatic, testicular, and rectal cancer [79,80].

To explore the influence of cholesterol on the orientation of KRas, we employed the definition of the orientation of KRas-4B-FMe described by Prakash et al. [20] to compare with the results from this work and propose a new method to define the orientation of KRas binding to the PM. In general, two order parameters have been adopted: (1) the distance (z) between C${}_{\alpha }$ atoms of the residue 132 on the lobe 2 and the residue 183 on the HVR, and (2) the angle Θ between the membrane normal direction and a vector running the C${}_{\alpha }$ atoms of the residue 5 and the residue 9 which belong to the first strand β1 in the structure of KRas-4B. The results of the density distribution of conformations defined by the order parameter *z* and cosΘ during the last 500 ns simulation time for four systems studied in this work have been presented in Figure 4.

Two distinct orientations of KRas were proposed in their work: OS${}_{1}$, in which the loop is solvent-accessible, and OS${}_{2}$, in which the effector-binding loop is occluded by the membrane. The remaining conformations are categorized into the intermediate state OS${}_{0}$.

Moreover, we define a new parameter, the angle Φ that runs the membrane normal and a vector running the C${}_{\alpha }$ atoms of residues 163 and 156 on the last helix α5 of lobe 2. According to data from references [81,82,83], it is known that dimerization of KRas-4B is a requirement for KRas signaling activity and tumor growth. The helix α5 has been reported to be involved in its dimerization interface. However, despite this being a relevant and interesting topic, dimerization of KRas-4B is outside of the scope of the MD study reported here, since the classical force field we have employed in the present work (CHARMM36) does not allow us to simulate the breaking and formation of chemical bonds. Due to its highly conserved structure for the CD, providing the information of the angle Θ along with the Φ could provide a new way for researchers to define the movement and orientation for KRas when binding to the membranes. We have calculated the angle and distance as described above. Moreover, the parameter we newly introduce here will be discussed further later.

Through the two-dimensional histogram, (z, cosΘ), three orientation states OS${}_{1}$, OS${}_{2}$, and OS${}_{0}$ were reported to be centered around (1.86, −0.5), (4.97, 1), and (3.33, 0.9), respectively, according to Prakash et al. [20]. In OS${}_{1}$, KRas-4B can interact with other proteins in cells, confirming that cholesterol has an important impact on the signaling activity for KRas-4B, especially for mutant ones, by increasing the flexibility and fluctuation in its CD with the exposed effector-binding loop. In Figure 4, we can observe that only when oncogenic KRas-4B-Far is bound to the chol.-30% membrane, can OS${}_{1}$ be shown for KRas-4B in a time span of 500 ns. As expected, when binding to the chol.-30% membrane, the wild-type Kras-4B-Far has been observed to stay in its inactive state (OS${}_{2}$). From the last 500 ns simulation time analyzed in this work, when binding to the anionic cholesterol-rich membrane, the wild-type and mutant KRas-4B-Far proteins can reach all three conformational regions, indicating more flexibility for the CD in the membrane normal direction and less affinity to the cholesterol-rich PM.

However, using these two coordinates (z, cosΘ) defined by Prakash et al. makes it difficult for us to categorize the conformational states of (wild-type and oncogenic) KRas-4B-Far for the cholesterol-free systems, and no clear OS${}_{1}$, OS${}_{2}$, and OS${}_{0}$ have been observed, see the upper panels in Figure 4. However, OS${}_{1}$ for the oncogenic KRas-4B-Far in Figure A3 and OS${}_{2}$ for the wild-type KRas-4B-Far in Figure A4 have been observed for the chol.-0% membrane systems. So, using two well-defined angles to describe the orientation of KRas-4B on the anionic membrane could be a good idea.

We have also investigated the time evolution of *z* for each system as reported in Figure A2, which shows major conformational fluctuations for four systems (oncogenic KRas-4B-Far and wild-type KRas-4B-Far, for 0 and 30% chol). For the cholesterol-free membrane systems, the protein majorly fluctuates between two distinct states in ranges of 2.4 ≤ *z* ≤ 4.3 and *z* > 4.3 nm, rarely visiting lower *z* values.

For the two cholesterol-free systems, are the orientation states of wild-type and oncogenic KRas-4B-Far always in its intermediate state OS${}_{0}$ according to Prakash et al.’s work [20]? This is a question that we want to answer.

### 2.5. Reorientation of Mutant Kras-4b-Far on the Anionic Membranes

By adopting the two angles (Θ and Φ) defined above, we analyzed the corresponding density profiles. We present the reorientation of the mutant KRas-4B-Far when bound to the anionic membrane with 30% of the cholesterol in Figure 5. Results for the remaining three systems studied here are reported in Figure A3, Figure A4 and Figure A5.

From Figure 5, the reorientation of mutant KRas-4B-Far on the chol.-30% bilayer has been observed during the 500 ns simulations time, giving a hint on the low free energy barriers between two orientation states (OS${}_{1}$-OS${}_{0}$, and OS${}_{0}$-OS${}_{2}$). Mutant KRas-4B spends most of the time in the active OS${}_{1}$ state, centered at (80${}^{\circ }$, 105${}^{\circ }$) on chol.-30% membrane, and fluctuates around (99${}^{\circ }$, 83${}^{\circ }$) when binding to the chol.-free bilayer, also in its OS${}_{1}$ state. Wild-type KRas-4B protein, regardless of the cholesterol’s content, prefers staying in its inactivate state, centered at (59${}^{\circ }$, 52.5${}^{\circ }$) and (53${}^{\circ }$, 37${}^{\circ }$) for chol.-0% and chol.-30%, respectively. This suggests that the orientation with the effector-binding loop occluded by the membrane (OS${}_{2}$) is disfavored in the wild-type KRas-4B-Far protein.

Here, we could conclude that high cholesterol in the PM helps KRas-4B mutant stay in its constitutively active state, which suggests that high cholesterol intake can increase mortality and may promote cancer progression for cancer patients. Our findings agree with the experimental and clinical results [55,56,58,59].

## 3. Methods

We performed four independent MD simulations of wild-type and mutated GTP-bound KRas-4B-Far attached to DOPC/DOPS (4:1) bilayers.

Eventually, some of the lipids were replaced by cholesterol molecules in such a way that two cholesterol percentages were considered: 0% and 30%. Each system contains a total of 304 lipid molecules fully solvated by 60,000 TIP3P water molecules and 48 potassium chloride at the human body concentration (0.15 M), yielding a system size of 222,000 atoms. All MD inputs were generated using a CHARMM-GUI web-based tool [84]. The force field was CHARMM36m for proteins [85] and CHARMM36 [86] for other molecules in each system. The crystal structure of KRas-4B with the partially disordered hypervariable region (pdb 5TB5) and GTP (pdb 5VQ2) were used to generate full-length GTP-bound KRas-4B-Far proteins. Two sequences of the wild-type and oncogenic KRas-4B-Far are presented in Figure A6.

After model building, each system was energy minimized for 5000 steps followed by three 250 ps simulations, and then four additional 500 ps equilibrium runs while gradually reducing the harmonic constraints on the systems. We used the NPT ensemble with the constant pressure of 1 atm maintained by the Parrinello–Rahman piston method with a damping coefficient of 5 ps${}^{-1}$ and temperature of 310.15 K controlled by the Nosé–Hoover thermostat method with a damping coefficient of 1 ps${}^{-1}$. Meaningful production runs were performed with an NPT ensemble for 1 μs from the last configuration of equilibrium run for each system, for a total of 4 μs. Time steps of 2 fs were used in all production simulations and the particle mesh Ewald method with a Coulomb radius of 1.2 nm was employed to compute long-ranged electrostatic interactions. The cutoff for Lennard–Jones interactions was set to 1.2 nm. In all MD simulations, the GROMACS/2018.3 package was employed [87] and periodic boundary conditions in three directions of space have been taken.

## 4. Conclusions

In this work, we performed MD simulations on four systems of wild-type/oncogenic KRas-4B-Far protein binding to membranes with different cholesterol contents (0% and 30%) to study the influence of cholesterol on the orientation of KRas. KRas-4B-Far shows the condensing effect on the area per lipid of the anionic model membrane through strong interactions between its CD and HVR moieties with the head groups of the lipids. More flexibility in its CD structure of KRas-4B-Far has been observed when binding to the PM with high cholesterol concentration, for both wild-type and mutant KRas-4B-Far proteins. The reorientation of mutant KRas-4B-Far on the anionic chol.-30% model membrane has been observed during the 500 ns simulations time, giving a hint on the low free energy barriers between a pair of orientation states (e.g., OS${}_{1}$-OS${}_{0}$, and OS${}_{0}$-OS${}_{2}$).

It has been shown for the first time that cholesterol makes it much easier for the mutant KRas-4B-Far shifting between different orientation states. The high cholesterol content in the PM favors OS${}_{1}$, exposing the effector-binding loop for signal transduction in cells from the atomic level. We propose that during the treatment of KRas-4B-related cancers, reducing the cholesterol level in the PM and sustaining cancer progression by controlling the plasma cholesterol intake should be taken into account in anti-cancer therapies. The present study of the role of cholesterol in Kras-4B orientation can provide one more direction and method for the treatment and prevention of cancer. By conducting four μs MD simulations, we confirm that high cholesterol in the PM helps KRas-4B mutant stays in its constitutively active state, which suggests that high cholesterol intake can increase mortality and may promote cancer progression for cancer patients.

## Figures and Tables

**Figure 1 membranes-10-00364-f001:**
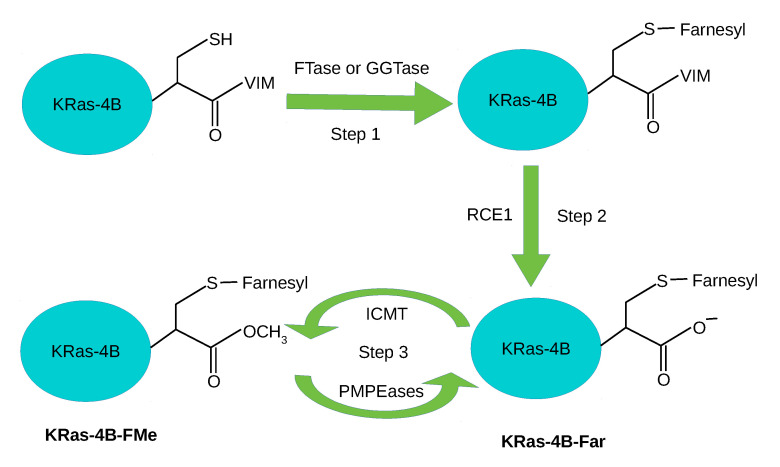
Post-translational modifications (PTMs) steps of KRas-4B: prenylation, hydrolysis, carboxymethylation and decarboxymethylation.

**Figure 2 membranes-10-00364-f002:**
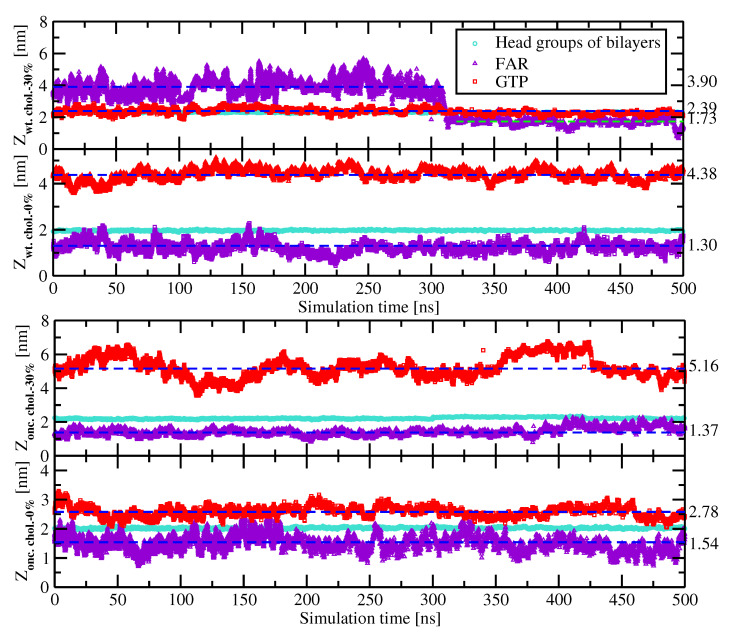
The localizations of the farnesyl group (FAR) and GTP of four KRas-4B systems studied in this work with respect to the center of the membrane along with the membrane normal as a function of simulation time. Geometric centers of the FAR, GTP, and phosphorus atoms of DOPC lipids from both leaflets are indicated as triangle up in violet and circle in turquoise, respectively. Data for chol.-30% shown here are adopted from our previous work [24] for the convenience of the audience. The average values of the FAR and GTP are indicated in dashed lines.

**Figure 3 membranes-10-00364-f003:**
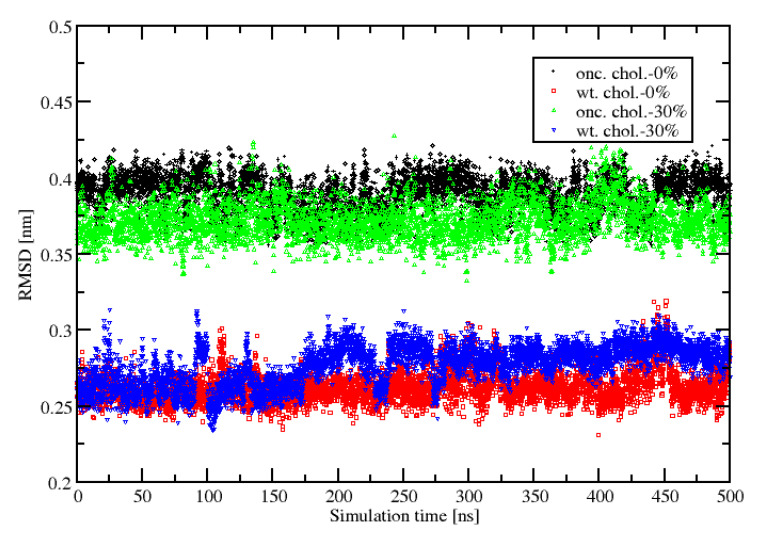
The RMSD of the 5-aa-sequence of the HVR during the last 500 ns of the 1 μs time span for four systems.

**Figure 4 membranes-10-00364-f004:**
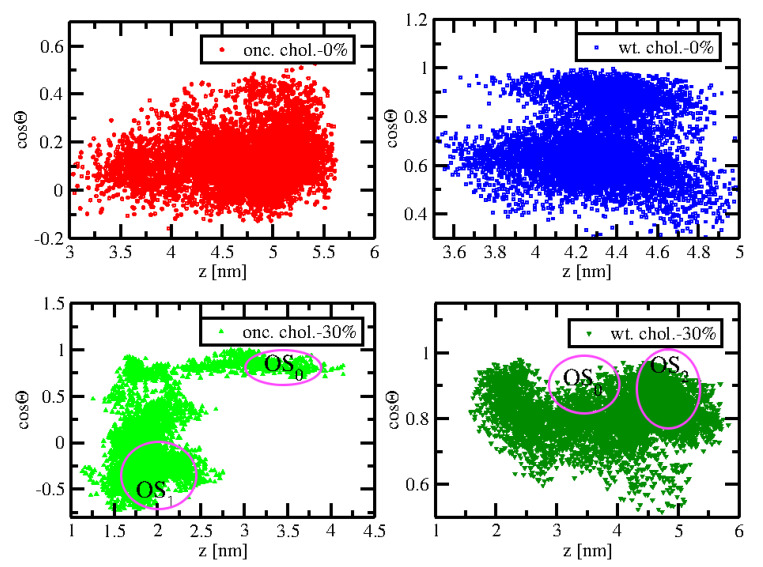
Density distributions of configurations of the oncogenic KRas-4B-Far and wild-type KRas-4B-Far systems with given values of coordinates *z* and cosΘ for 0 and 30% cholesterol. Observed OS${}_{1}$, OS${}_{2}$, and OS${}_{0}$ have been encircled in their corresponding locations.

**Figure 5 membranes-10-00364-f005:**
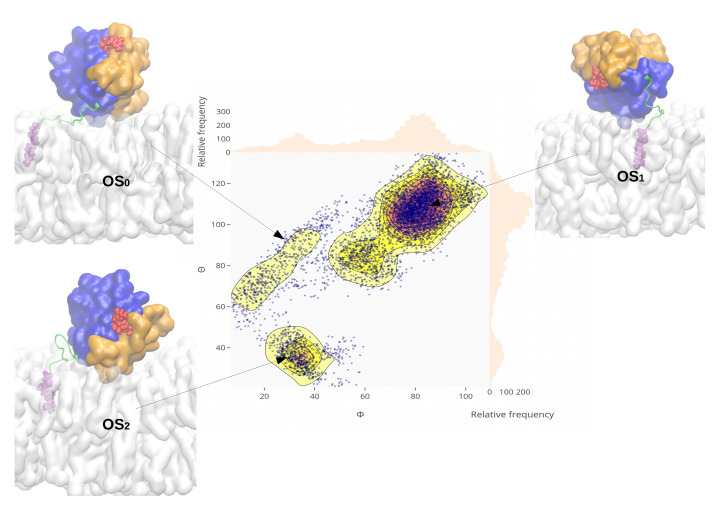
Reorientation of the oncogenic KRas-4B on the chol.-30% membrane. Density distribution of conformations projected onto a plane defined by the reaction coordinates Φ and Θ in degrees (${}^{\circ }$). The relative frequency of each coordinate is shown on the right and upsides. The membrane bilayer is shown as a white surface. Lobe 1 is highlighted in orange, lobe 2 in blue, HVR backbone in green, GTP in red, and FAR in violet. Water and ions are not shown here for the sake of clarity.

**Table 1 membranes-10-00364-t001:** The average area per lipid (A) and thickness (Δz) of the anionic membrane for four KRas-4B-Far systems studied in this work. The thickness of the membrane Δz by computing the mean distance between phosphorus atoms of DOPC head groups from both leaflets. Estimated errors in parenthesis.

System	A (nm${}^{2}$)	Δz (nm)
wt. chol.-0%	0.679 (0.008)	3.84 (0.04)
wt. chol.-30% [24]	0.523 (0.007)	4.35 (0.05)
onc. chol.-0%	0.679 (0.008)	3.89 (0.04)
onc. chol.-30% [24]	0.525 (0.006)	4.23 (0.04)

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
