# Peer review of "Influence of Cholesterol on the Orientation of the Farnesylated GTP-Bound KRas-4B Binding with Anionic Model Membranes"

_membranes, 2020, doi:10.3390/membranes10110364_

Round 1

Reviewer 1 Report

In this manuscript, Martí et al describe the effect of cholesterol content on wild-type and oncogenic Krass-4B-Far protein binding to the plasma membrane. The results indicate a cholesterol content of 30% in the plasma membrane favors the active orientation state of mutant KRas-4B, suggesting cholesterol potentially promotes KRas constitutive activation. Overall the data is presented clearly and the authors provide validation studies for their MD simulations; however, the manuscript would benefit from the following minor improvements.

  1. The reason for selecting 30% cholesterol for the MD simulations is unclear from the text. Please describe the experimental or biological reason for 30% cholesterol PM content. Further, is this percentage comparable to the membrane cholesterol composition in healthy adults or hypercholesterolemia patients?
  2. Minor grammatical error should be corrected, specifically run on sentences, and abbreviations need to be spelled out when first appearing in text. For example, PM and MD in the abstract.

Author Response

Dear prof. Reviewer 1,

Good morning. Here is my point-by-point response to your valuable comments, please have a look.

Point 1: The reason for selecting 30% cholesterol for the MD simulations is unclear from the text. Please describe the experimental or biological reason for 30% cholesterol PM content. Further, is this percentage comparable to the membrane cholesterol composition in healthy adults or hypercholesterolemia patients?

Response 1: It is a good suggestion. In different types of cells, the content of cholesterol is different. We have introduced different percentages of cholesterol in different types of cell membranes, shown in lines 115-121. And cholesterol constitutes about 33.3% of the outer leaflet in healthy colorectal cells has been provided, which is in good agreement with 30% of cholesterol adopted in this work.

And the reason why we choose 30% cholesterol is explained in lines 121-123.

Point 2: Minor grammatical error should be corrected, specifically run on sentences, and abbreviations need to be spelled out when first appearing in text. For example, PM and MD in the abstract.

Response 2: We have gone through typo and grammar check for two times and PM and MD in the abstract have been spelled out.

Yours sincerely,

Huixia Lu, Ph.D.

Department of Physics,

Technical University of Catalonia-Barcelona Tech,

B4-210 North Campus UPC,

08034 Barcelona, Catalonia (Spain)

email: huixia.lu@upc.edu

Reviewer 2 Report

Authors present molecular dynamic simulation data on the orientation of farnesylated GTP-bound wild type or oncogenic KRas-4b using anionic model membranes. Based on area per lipid calculations and radial distribution functions authors confirm that high cholesterol concentrations may help oncogenic KRas-4b stay in the constitutive active state. Authors conclude that reduction in cholesterol levels in treatments of cancer patients should be considered.

Major points:

  1. Experimental data, e.g. on nanodiscs or on life cells are missing in this manuscript to support the theoretical data shown here.

2. The manuscript is hard to read!

i) Figures legends are unclear: „......Figure 4: Density distribution of conformations defined by the order parameter z and cosÏ´.“  There are no descriptions given as legend to Figure 4, at all.

In the relevant text lines 215, 216 authors write „ ……..Time evolution of z for each system is reported in Fig. A2, which shows major conformational fluctuations for four systems……….”  Looking up Fig A2 I find MD simulation of GTP-bound G12D-PHOS-KRas-Far. However, in the subsequent text lines 216, 217 authors continue to write „……… From the last 500 ns simulation time analyzed in this work, wild-type and mutant KRas-4B-Far proteins could reach all three conformational regions……“

Question: Which molecules were used for the calculations of conformational fluctuations in Figure 4:  G12D-PHOS-KRas-Far or KRas-4B-Far or others? There should be a large difference for the conformational fluctuations between phosphorylated and non –phosphorylated molecules?

ii) Authors define an angle between the membrane normal and a vector running the Cα atoms of residues 163 and 156 on the last helix α5. The α4-β6-α5 region of RAS (including helix α5) is involved in the dimerization of KRas as described in several publications (see Spencer-Smith, R., Koide, A., Zhou, Y. et al. Inhibition of RAS function through targeting an allosteric regulatory site. Nat Chem Biol 13, 62–68 (2017) https://doi.org/10.1038/nchembio.2231. Ambrogio et al. 2017 https://doi.org/10.1016/j.cell.2017.12.020; or Khan, I., Spencer-Smith, R. & O’Bryan, J.P. Targeting the α4–α5 dimerization interface of K-RAS inhibits tumor formation in vivo. Oncogene 38, 2984–2993 (2019). https://doi.org/10.1038/s41388-018-0636-y). As the dimerization of KRas is a requirement for KRas signaling activity and tumor growth, the topic of KRAS dimerization needs to be incorporated into the context of conformations discussed here in the presence and absence of cholesterol.

-Authors definitions refer to membrane orientations OS1,OS2 and OS0 as proposed in the paper by Prakash et al. 2019. For the diagrams shown in Fig 4 authors should explain why certain regions in their conformational fluctuations correlate with OS1-OS0 and OS0-OS2. They could even encircle these OS regions in all four diagrams of Fig 4 within their conformational fluctuations defined by order parameters z and cosÏ´?

Minor points

In the „Introduction“ the text is partially difficult to understand:  lines 88: „......Chain of the amino acid serine, then the phosphorylated serine (PHOS) is obtained……….“ Part of the sentence is missing?

Figure 2: The letters to describe the ordinates are much too small to be read.

Explain abbreviation „RMSD“

Author Response

Dear prof. Reviewer 2,

Good morning, here is my point-by-point response to your valuable comments, please have a look.

Point 1: Experimental data, e.g. on nanodiscs or on life cells are missing in this manuscript to support the theoretical data shown here.

Response 1: We deeply thank you for your kind suggestion which helps us improve the text. Experimental data has been provided in lines 128-135.

Point 2:

i) Figures legends are unclear: „......Figure 4: Density distribution of conformations defined by the order parameter z and cosÏ´.“ There are no descriptions given as legend to Figure 4, at all. In the relevant text lines 215, 216 authors write „ ……..Time evolution of z for each system is reported in Fig. A2, which shows major conformational fluctuations for four systems……….” Looking up Fig A2 I find MD simulation of GTP-bound G12D-PHOS-KRas-Far. However, in the subsequent text lines 216, 217 authors continue to write „……… From the last 500 ns simulation time analyzed in this work, wild-type and mutant KRas-4B-Far proteins could reach all three conformational regions……“

Question: Which molecules were used for the calculations of conformational fluctuations in Figure 4: G12D-PHOS-KRas-Far or KRas-4B-Far or others? There should be a large difference for the conformational fluctuations between phosphorylated and non-phosphorylated molecules?

Response 2-i): We apologize for this misleading expression in our original text. GTP-bound G12D-PHOS-Kras-Far has been changed into “the onc. and wt. Kras-4B-Far proteins for four cases studied in this work”. We have only studied the wild-type Kras-4B-Far and the mutated Kras-4B-Far (mutation G12D and PHOS at the site Ser-181 were applied). And the addition of the active phosphate (PO34−) group to the side chain of the amino acid serine will highly influence the conformational fluctuation of Kras-4B.

Furthermore, the description for Fig. 4 can be found in line 222 and later on in our first submission, which is a little far away from Fig. 4. In order to avoid misapprehension, we have moved this paragraph under Fig. 4.

ii) Authors define an angle between the membrane normal and a vector running the Cα atoms of residues 163 and 156 on the last helix α5. The α4-β6-α5 region of RAS (including helix α5) is involved in the dimerization of KRas as described in several publications (see Spencer-Smith, R., Koide, A., Zhou, Y. et al. Inhibition of RAS function through targeting an allosteric regulatory site. Nat Chem Biol 13, 62–68 (2017) https://doi.org/10.1038/nchembio.2231. Ambrogio et al. 2017 https://doi.org/10.1016/j.cell.2017.12.020; or Khan, I., Spencer-Smith, R. & O’Bryan, J.P. Targeting the α4–α5 dimerization interface of K-RAS inhibits tumor formation in vivo. Oncogene 38, 2984–2993 (2019). https://doi.org/10.1038/s41388-018-0636-y). As the dimerization of KRas is a requirement for KRas signaling activity and tumor growth, the topic of KRAS dimerization needs to be incorporated into the context of conformations discussed here in the presence and absence of cholesterol.

-Authors definitions refer to membrane orientations OS1,OS2 and OS0 as proposed in the paper by Prakash et al. 2019. For the diagrams shown in Fig 4 authors should explain why certain regions in their conformational fluctuations correlate with OS1-OS0 and OS0-OS2. They could even encircle these OS regions in all four diagrams of Fig. 4 within their conformational fluctuations defined by order parameters z and cosÏ´?

Response 2-ii): We have added three refs suggested here in lines 233-235 and 241-243, however, we also mention that “dimerization of KRas-4B is outside of the scope of the MD study reported here since the classical force field we have employed in the present work (CHARMM36) does not allow to simulate the breaking and formation of chemical bonds.”

We have encircled the locations of OS1, OS2, and OS0 in Fig. 4, correspondingly description and explanation in the text has been added, see lines 256-267.

point 3: Minor points

In the „Introduction“ the text is partially difficult to understand: lines 88: „......Chain of the amino acid serine, then the phosphorylated serine (PHOS) is obtained……….“ Part of the sentence is missing?

Response 3: Mistake of this broken sentence has been corrected, see lines 87-89.

Point 4: The letters to describe the ordinates are much too small to be read.

Response 4: Fig. 2 has been exaggerated and its ordinates are changed into bold style.

Point 5: Explain abbreviation „RMSD“

Response 5: Abbreviations, such as MD, COSMIC, RMSD, PDEδ, Ftase, GGTase, RCE1, ICMT, PMPEases, TSPO, and SI have been well-explained.

Yours sincerely,

Huixia Lu, Ph.D.

Department of Physics,

Technical University of Catalonia-Barcelona Tech,

B4-210 North Campus UPC,

08034 Barcelona, Catalonia (Spain)

email: huixia.lu@upc.edu

Round 2

Reviewer 2 Report

The points raised were partially answered.

Point 1: The authors have just added text reviewing experimental data by others but own experimentall data are still missing.

Point 2i) In their respose letter authors define that they have used for their simulations GTP-bound G12D-PHOS-Kras4B-Far with phophorylation on serine 181. However, I could not find this definition in the present version of the  manuscript. Authors should clearly state for the reader that all data given in this manuscript are based on Kras4B with phosphorylation at serine 182 and not at serine 171, unless mentioned otherwise.

Explicitly mentioning again Ser-181 in the legend to Figure A6 should be avoided in case that G12D-PHOS-Kras-Far with phosphorylated Ser181 was used throughout the study.

Point 2ii) The putative dimerization interface  (α4-β6-α5 region) of KRAS is available from the references mentioned and includes a region around R135. Could you estimate from your density distributions of configurations if the putative dimerization interface is exposed to allow dimerisazion?

Minor point:  Line 211 : „…… two mutations in the sequence of KRas-4B on the same type of KRas-4B-Far…..“. Question: G12V is one mutation. Which additional mutation do you refer to?

Author Response

Dear Reviewer 2:

Thanks for your valuable comments, my answers are as follows:

Point 1: The authors have just added text reviewing experimental data by others but own experimental data are still missing.

Response 1: We would like to establish that we are a computational, not an experimental group so that we don’t have our own experimental results.

Point 2i): In their respose letter authors define that they have used for their simulations GTP-bound G12D-PHOS-Kras4B-Far with phophorylation on serine 181. However, I could not find this definition in the present version of the manuscript. Authors should clearly state for the reader that all data given in this manuscript are based on Kras4B with phosphorylation at serine 182 and not at serine 171, unless mentioned otherwise.

Explicitly mentioning again Ser-181 in the legend to Figure A6 should be avoided in case that G12D-PHOS-Kras-Far with phosphorylated Ser181 was used throughout the study.

Response 2i): The definition of PHOS has been shown to lines 89-90, and phosphorylation on the site-171 is not considered in this work. And re-definition of PHOS in the legend to Fig. A6 has been avoided.

Point 2ii) The putative dimerization interface (α4-β6-α5 region) of KRAS is available from the references mentioned and includes a region around R135. Could you estimate from your density distributions of configurations if the putative dimerization interface is exposed to allow dimerisazion?

Response 2ii): The main goal of our paper is to identify the orientational distributions of KRas-4B under the influence of cholesterol when binding to the DOPC/DOPS(4:1). Of course, the distribution of the dimerization interface (α4-β6-α5 region) of KRAS would be very interesting. To do this kind of study we would need to run new simulations where the dimerization interface is targetted in a meaningful way (order of 1 microsecond) and this is out of the scope of the present work. This has been pointed out on page 8 (line 244) of the manuscript.

Point 3: Minor points

Line 211 : „…… two mutations in the sequence of KRas-4B on the same type of KRas-4B-Far…..“. Question: G12V is one mutation. Which additional mutation do you refer to?

Response: two mutations have been explained in lines 212-214.

In biology, an alteration in the nucleotide sequence of can be defined as a mutation, like G12D adopted in our work. Spontaneous mutations can be characterized by the specific change, for instance, tautomerism, depurination, deamination, etc. (see https://en.wikipedia.org/wiki/Mutation). We believe that phosphorylation can also be considered as a mutation in the structure of the site of Ser-181.

We did not consider G12V in our work.

Yours sincerely,

Huixia Lu, Ph.D.

email: huixia.lu@upc.edu